

# Deposition velocity concept does not apply to fluxes of ambient aerosols

Rostislav Kouznetsov[1], Mikhail Sofiev[1], Andreas Uppstu[1], and Risto Hänninen[1]

[1]Finnish Meteorological Institute, Helsinki, Finland

**Correspondence:** Rostislav Kouznetsov (Rostislav.Kouznetsov@fmi.fi)

**Abstract.** The process of dry deposition in chemistry-transport models is usually implemented assuming a proportionality between the deposition flux and the corresponding concentration of a tracer at some reference height. The coefficient of proportionality, called deposition velocity, $V_d$, is to be parameterized and validated experimentally. We analyse large discrepancies between field and wind-tunnel measurements of $V_d$ of aerosols with aerodynamic diameters ranging from approximately $0.1\mu m$ to $2\mu m$. In seemingly similar conditions, the deposition velocities reported in different experiments may differ by up to two orders of magnitude, with field measurements showing much higher values than experiments performed in controlled environments with known particle properties. We demonstrate that the bulk of the discrepancy can be explained by fast chemical reactions and a particle-to-gas conversion in the immediate vicinity of the surface. By applying the chemistry-transport model SILAM, equipped with gas-particle partitioning for ammonium nitrate, we demonstrate that in the presence of even small amounts of ammonium nitrate, the vertical flux of total aerosol mass is not controlled by particle deposition but rather by aerosol-gas partitioning in the vicinity of the surface. While there are many other non-conservative components in ambient aerosols apart from ammonium nitrate, we demonstrate that the abundance of ammonium nitrate alone is sufficient to render typical ambient aerosol into a non-conservative substance. Under these conditions, the deposition flux is not proportional to the concentration, and the concept of deposition velocity as a proportionality coefficient between concentration and deposition flux falls apart. By simulating a renowned field experiment with the SILAM model, we are able to reproduce the magnitudes and temporal behaviors of ambient particle fluxes using the deposition parameterization derived from wind-tunnel studies.

## 1 Introduction

Dry deposition is one of the key mechanisms for removal of gaseous and particulate species from the atmosphere. The measure of dry deposition is the flux $F$ of a substance, expressed as amount (e.g. mass or number) deposited per unit area per unit time. In many applications, the flux is *assumed* to be proportional to the substance concentration $C$ at some reference height above the surface:

$$F = V_d(\text{substance}, \text{flow}, \text{surface}) \cdot C, \tag{1}$$

where $V_d$ is a proportionality coefficient of a velocity dimension, called the deposition velocity. The assumption implies that $V_d$ can be a function of flow, surface, and substance properties, but not a function of $C$.



All dry deposition schemes for particles used in atmospheric dispersion models that we are aware of can be expressed in the form of Eq. (1), (e.g. Slinn and Slinn, 1980; Giorgi, 1986; Zhang et al., 2001; Kouznetsov and Sofiev, 2012; Zhang and He, 2014; Pleim et al., 2022). Eq. (1) follows from the basic law of mass conservation (Businger, 1986) if one assumes: (1) a quasi-steady-state process, (2) horizontal homogeneity at a sufficient scale, (3) absence of sources and sinks of the substance between the reference height and the surface, and (4) zero concentration *at* the surface. The assumptions (1)-(3) taken together

are equivalent to the constant-flux assumption, i.e. the flux is constant along the vertical between the reference height and the surface (Sofiev, 2002; Kouznetsov and Sofiev, 2012). The assumption (4) implies that particles, once reach the surface, get captured regardless the presence of other particles. If any of these assumptions is violated, the flux proportionality to the concentration at the reference height does not hold, and the concept of deposition velocity becomes inapplicable.

    Along with the definition given by Eq. (1), a wider definition of deposition velocity has been adopted within the flux

measurement community (Vong et al., 2004, 2010; Grönholm et al., 2009; Matsuda et al., 2010; Sun et al., 2014; Deventer et al., 2015; Petroff et al., 2018). The deposition velocity of a substance is defined as the vertical flux $F$ of the substance at a reference height above the surface, normalized with its concentration $C$ at the same height, i.e.

$$V_{d,a} = F/C. \tag{2}$$

$V_{d,a}$ is thus a feature of each individual measurement, and computed from the two observed quantities. Moreover, its inde-

pendence from the concentration postulated in the Eq. (1) is not declared. In the following, we will use the term "apparent deposition velocity" for the one defined by Eq. (2) to distinguish it from the definition given by Eq. (1). These two definitions are equivalent only if all four assumptions behind Eq. (1) hold.

    Both definitions appeared together for the first time probably in the review of Sehmel (1980), however no explicit distinction between them has been done there. An extensive review of Pryor et al. (2008) adopts definition (2), then mentions the impor-

tance of negligibility of storage and phase changes below the observation heights. Nemitz et al. (2009) used definition similar to (1) and explicitly mentioned the requirement of absence of chemical reactions, but in the next sentence $V_d$ was calculated using the average deposition velocity as defined in (2). A recent review of Farmer et al. (2021) uses the definition in form of (1) and mentions that $V_d$ is independent from ambient concentration, but without explicitly specifying that this independence is based on *assumptions* and not guaranteed for arbitrary particles. A few experimental studies (Rohbock, 1982; Hicks et al.,

1987) clearly mention the requirement of proportionality between the flux and concentration therefore stick to Eq. (1). A substantial number of experimental studies (Sievering et al., 1981; Sievering, 1981; Rannik et al., 2003b; Mammarella et al., 2011; Zhang and He, 2014; Matsuda et al., 2015; Lavi et al., 2013; Connan et al., 2018) address deposition velocity without giving an explicit definition, and proportionality between flux and concentration was probably assumed in some of them, but obviously not considered in some others.

Many studies have been conducted to infer deposition velocities for different airborne substances in gaseous and aerosol forms under both laboratory and field conditions. They can be classified into two categories: those where the deposition flux is calculated from the amount deposited *onto the surface* (direct method applying Eq. 1), and those where the flux is inferred *above the surface* from gradient or eddy-covariance measurements (indirect method). The direct method is more laborious and





requires a well-defined and easily identifiable substance, and it is therefore only applicable in laboratory conditions. While
the indirect method can be applied also outdoors, ensuring that assumptions (1)-(4) hold for the ambient aerosol mixture is
non-trivial.

Literature values for deposition velocities of aerosols in the accumulation-mode size range, i.e. from $0.1\,\mu m$ up to a few
micrometers, exhibit a large scatter, which has been puzzling researchers already since several decades (Garland, 2001). Studies
inferring deposition fluxes from direct measurements (Chamberlain, 1968; Clough, 1975; Zufall et al., 1998; Caffrey et al.,
1998) show that typical values of accumulation mode aerosol deposition velocities onto a smooth surface are of the order
of $10^{-4}\,\mathrm{m/s}$. These studies were performed in wind tunnels for both smooth solid and water surfaces, and for surfaces with
small roughness elements. Deposition velocities of the same order of magnitude were also found in the study by Sehmel and
Sutter (1974), who used a surrogate surface to collect material deposited on a natural lake. On the other hand, field studies
that apply flux or gradient methods above water surfaces have reported two orders of magnitude higher deposition velocities
of about $10^{-2}\,\mathrm{m/s}$ (Sievering et al., 1981; Zhang and He, 2014). Moreover, observations over snow or low vegetation show a
scatter of about two orders of magnitude. While some of them (Duan et al., 1988; Nemitz et al., 2002) agree with wind-tunnel
measurements, others (Sievering et al., 1981; Vong et al., 2010) indicate 1–2 orders of magnitude higher velocities.

Indirect measurements of particle deposition onto high vegetation have yielded deposition velocities of $10^{-2}\,\mathrm{m/s}$, with
little dependence on the particle size within a range of $0.2\,\mu m$ to $5\,\mu m$, as summarized by (Pleim et al., 2022). These deposition
velocities are two orders of magnitude higher than those predicted by mechanistic models (Slinn, 1982; Kouznetsov and Sofiev,
2012). To the best of our knowledge, there are no direct measurements of $V_d$ onto high vegetation.

Many of the original studies considered by Pleim et al. (2022) report strong temporal variability and even a change of
direction of the particle fluxes over high vegetation. The reported fraction of upward particle fluxes over forest ecosystems
range from 30% to 60% (Ahlm et al., 2010; Rannik et al., 2003a; Pryor et al., 2007, 2009; Gordon et al., 2011; Lavi et al.,
2013; Deventer et al., 2015). For this reason many of these studies avoid the term "deposition velocity" in favour of "transfer
velocity", while still calling the downward particle flux "deposition".

Several mechanisms causing such discrepancies have have been suggested to explain the upward fluxes. Nemitz and Sutton
(2004) and Farmer et al. (2013) pointed out that vertical fluxes of ambient aerosols can be substantially affected by $NH_4NO_3$
formation and decomposition due to the $NH_3 + HNO_3 \leftrightarrow NH_4NO_3$ equilibrium. Among other mechanisms breaking the par-
ticle mass conservation are interactions between aerosol particles (coagulation, agglomeration, fragmentation) or aerosol parti-
cles and the carrying gas (mass transfer of water vapour, condensation or evaporation of semi-volatile compounds), etc. (Farmer
et al., 2013; Petroff and Zhang, 2010). If any of these mechanisms impacts the vertical flux, it is no longer proportional to the
concentration, rendering Eq. (1) inapplicable. This discrepancy has been recognised in many deposition flux studies (e.g.
Farmer et al., 2013). The role of ammonium nitrate in particle flux formation has been studied in details by Nemitz et al.
(2009). However, particle concentrations dominated by non-volatile ammonium sulfate were considered sufficient justification
to neglect gas-particle partitioning (Petroff and Zhang, 2010). Therefore, apparent deposition velocities reported in field studies
have been used to develop parametrisations for particle deposition used in atmospheric transport models (Zhang et al., 2001;
Petroff and Zhang, 2010; Zhang and He, 2014; Emerson et al., 2020; Pleim et al., 2022). As a result, the obtained dry deposition



schemes predict $1 - 2$ orders of magnitude higher deposition fluxes than mechanistic models based on "first-principles" (Slinn, 1982; Kouznetsov and Sofiev, 2012).

The goal of the present study is to explore the discrepancy between different experimental studies of $V_d$. We show that the discrepancy does not originate from the natural variability of the particle-surface interaction, but rather from the inconsistency between the deposition velocity defined by Eq. (1) and the apparent deposition velocity defined by Eq. (2). While we primarily focus on the deposition of $NH_4NO_3$, our findings should be valid also for other non-inert aerosols. We use a chemistry-transport model SILAM that applies a first-principles based deposition scheme and accounts explicitly for relevant aerosol processes to simulate fluxes and depositions observed in one of the measurement campaigns. This allows us to directly identify the processes responsible for observed particle fluxes (Sec. 3). Then, by using a single-column setup of the model for a system undergoing the reaction $NH_3 + HNO_3 \leftrightarrow NH_4NO_3$, we demonstrate the difference between the particle deposition process and the processes responsible for particle fluxes above the surface (Sec. 4). Finally, we estimate the amount of in-air $NH_4NO_3$ needed to break the linear relationship between flux and concentration, and suggest an approach to bridge the gap between observed apparent deposition velocities and deposition parametrisations based on Eq. (1).

## 2 Methods

To demonstrate the ability of a chemistry-transport model to reproduce the observed range of apparent deposition velocities, we use the Eulerian chemistry transport model SILAM (System for Integrated modelLing of Atmospheric coMpostion, http://silam.fmi.fi, accessed 26.04.2025). The model features mass-conservative transport schemes (Sofiev, 2002; Sofiev et al., 2015), and a mechanistic particle deposition scheme derived from first principles (Kouznetsov and Sofiev, 2012). It also applies a scheme for secondary inorganic aerosol formation and gas-particle equilibrium that is capable of bi-directional gas-particle partitioning of ammonium nitrate with ammonia and nitric acid (Galperin and Sofiev, 1998; Sofiev, 2000). $NH_4NO_3$ is formed when $NH_3$ and $HNO_3$ are in abundance, and it decomposes back to the gaseous constituents when the product of their partial pressures drops below a temperature-dependent threshold (Mozurkewich, 1993). SILAM does not consider secondary emission of once-deposited $NH_3$, and is thus not expected to reproduce the upward flux of $NH_3$ or $NH_4NO_3$. The model has been extensively used and evaluated in numerous applications of air quality (Kukkonen et al., 2012; Sofiev et al., 2018; Petersen et al., 2019; Blechschmidt et al., 2020; Pachón et al., 2024; Colette et al., 2024), atmospheric composition (Kouznetsov et al., 2020; Sofiev et al., 2020) and emergency response (Brenot et al., 2021; Jylhä et al., 2018).

For the present study, we have performed two simulations with SILAM. The first one was a simulation of a field campaign by Gallagher et al. (1997) at the Speulder forest site in the Netherlands on June 29-30, 1993. The simulations included regular atmospheric pollutants, with the model setup being similar to that of Copernicus Atmosphere Monitoring Service (CAMS) operational forecasts of SILAM https://regional.atmosphere.copernicus.eu/ (accessed 26.04.2025). The simulation was driven by ECMWF ERA5 meteorology and nested into global SILAM simulations for the corresponding period. The European model setup generally followed the CAMS configuration (Colette et al., 2024). The global simulations closely followed the corresponding operational setup (Sofiev et al., 2020). From the results of the regional run, we extracted concentrations and deposi-



tions of $PM_{2.5}$, $NH_3$, $HNO_3$, $NH_4NO_3$ for the site location and the time period of the measurements performed by Gallagher et al. (1997), allowing a direct comparison of modelled and measured time series.

The second simulation applied an idealized single-column setup to explore the difference between the particle fluxes at the
reference height and the actual deposition fluxes at the surface. We simulated the 1D case with the disabled regular transport and wet deposition processes, thus keeping only gas-particle partitioning, vertical diffusion, settling of particles, and dry deposition. The simulation had a vertical resolution of 1m close to the surface, to enable explicit evaluation of profiles of vertical fluxes and actual deposition. Since the SILAM deposition model for smooth and water surfaces (Kouznetsov and Sofiev, 2012) has no tuning parameters, the simulations were performed over water picking a location in Northern Sea (56N,0E). The simulation was
driven with the hourly meteorological fields of ERA5 (vertical profiles of wind, temperature and humidity). The meteorological fields were used to derive parameters for dry deposition, vertical turbulent diffusion, and gas-particle equilibrium. The domain extended up to 250m above the surface, at which height a zero-flux boundary condition was enforced.

The $NH_4NO_3$ aerosol was simulated as a single bin with 0.7μm of dry diameter. Local equilibrium between $NH_3$, $HNO_3$, and $NH_4NO_3$ was enforced at every model time step. To separate diffusion- and chemistry-driven fluxes, we introduced a
gaseous chemically-inert substance with diffusion characteristics equivalent to those of $SO_2$. For such a gas, the constant-flux assumption is fulfilled in 1D simulations. Then the aerodynamic resistance between two adjacent layers could be evaluated from the deposition flux of the pseudo-$SO_2$ and the difference in its concentrations between the levels. Having the aerodynamic resistance, one can evaluate the diffusion-driven vertical flux of all species from gradients of their concentrations at the corresponding levels.

Since the model uses a process split, it is important to make sure that the order of the processes and the time step of the simulations does not affect the results. To confirm it, we have performed a series of the simulations with time steps from 5 seconds to 10 minutes. The impact of the time step on the concentrations of the species of interest becomes noticeable at time steps longer than one minute.

## 3  Simulating particle fluxes under real-life conditions

The particle deposition scheme in SILAM (Kouznetsov and Sofiev, 2012) predicts deposition velocities for sub-micron particles of a fraction of a millimeter per second. At the same time, other schemes fitted with the data of field experments (Zhang et al., 2001; Emerson et al., 2020; Pleim et al., 2022) predict much higher deposition velocities, $\sim 1\ \mathrm{cm s^{-1}}$.

Figure 1a shows a sketch[1] of the particle-fluxes time series observed by Gallagher et al. (1997) above a coniferous forest at the Speulder site in the Netherlands on June 29–30, 1993. Other panels show the concentrations and deposition fluxes of the
SILAM simulation for the same time period and location.

The simulated time series of $PM_{2.5}$ and $NH_4NO_3$ indicate that during the considered period, concentrations of $NH_3$ and $HNO_3$ were high enough to form particulate $NH_4NO_3$. Since the deposition rates of both gases are much higher than that

---

[1]The original paper is copyrighted by the Elsevier Group, which refused to give a permission to use their intellectual property in the openly-distributable publication. Readers are encouraged to reach out to the paper and compare our results to the original figure.





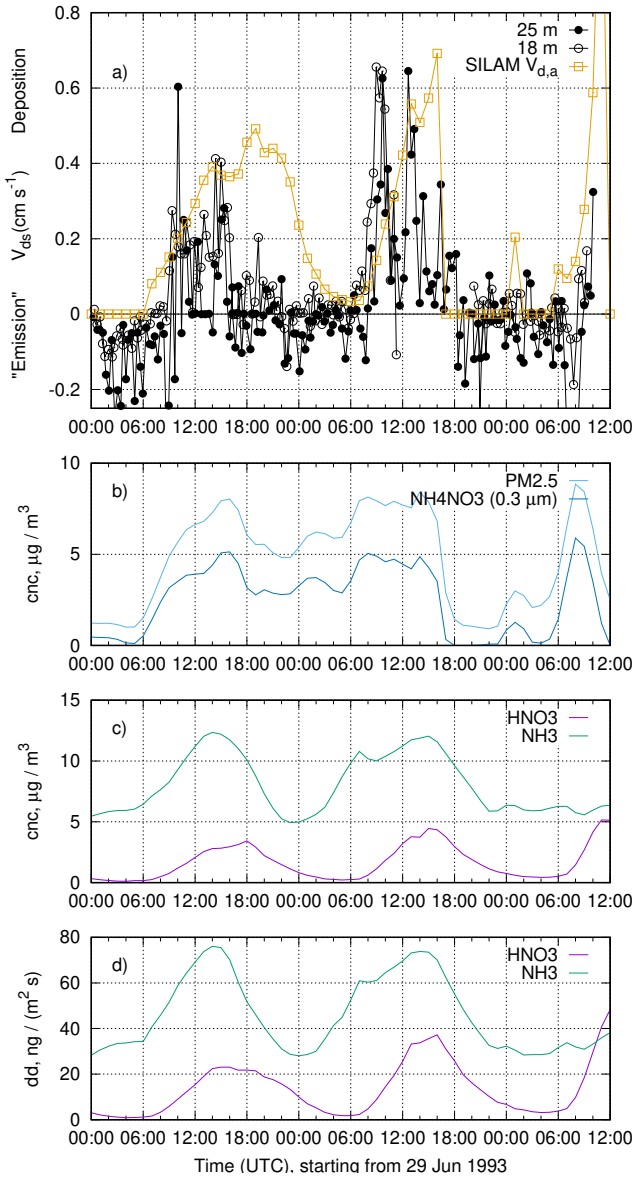

**Figure 1.** Observed apparent deposition velocity and simulated concentrations and deposition fluxes of the key species in the campaign of Gallagher et al. (1997). $V_{ds}$ in the original paper stands for the apparent deposition velocity compensated for aerodynamic resistance. For our purpose the difference between $V_{ds}$ and $V_{d,a}$ is negligible. Panel a: a sketch of the timeseries of aerosol fluxes observed by Gallagher et al. (1997, Fig. 1 there) with an overlay of the apparent deposition velocity derived from SILAM, panel b: SILAM-predicted concentrations of $NH_4NO_3$ and total $PM_{2.5}$; panel c: SILAM-predicted concentrations of $HNO_3$ and $NH_3$; panel d: SILAM-predicted dry deposition fluxes of $HNO_3$ and $NH_3$.





of $NH_4NO_3$, the downward particle flux was largely dominated by decomposition of $NH_4NO_3$ near the surface and the subsequent deposition of $NH_3$ and $HNO_3$. This very process was hypothesised to be the reason for the observed fluxes by

Nemitz and Sutton (2004). The periods of observed strong aerosol "deposition" fluxes closely correspond to periods when the concentrations of $NH_4NO_3$ and both gases were high in the model. Under these conditions, the modelled deposition flux is controlled by intensively depositing $HNO_3$. The resulting near-surface deficit of $HNO_3$ shifts the gas-particle fractionation and causes disintegration of $NH_4NO_3$, which, in turn, creates a net downward flux of the aerosol well comparable with the observed fluxes.

For a comparison, we have plotted an apparent deposition velocity $V_{d,a}$ deduced from the simulation (Figure 1a) as if the total particle flux was fully controlled by the deposition of $HNO_3$. This apparent deposition velocity can be deduced from concentration of $C_{PM2.5}$ and the dry-deposition flux $D_{HNO3}$ at the surface reported by the model.

$$V_{d,a} = \begin{cases} 0, & \text{if } C_{NH4NO3} < 1\mu gm^{-3} \text{ ;} \\ \dfrac{M_{NH4NO3}}{M_{HNO3}} \dfrac{D_{HNO3}}{C_{PM2.5}}, & \text{otherwise.} \end{cases} \qquad (3)$$

The ratio of molar masses $M$ is needed to convert the deposition mass flux of $HNO_3$ to the equivalent mass flux of $NH_4NO_3$.

We have chosen a concentration threshold for the $NH_4NO_3$ of $1\mu gm^{-3}$ as a criterion of abundance of particulate phase. Without this threshold the division of small values caused excessive noise in the time series. The observed triple peak pattern and the magnitude of the simulated $V_{d,a}$ timeseries is well reproduced by the model. The imperfect timing of the peaks can be attributed to the uncertainties in the model, imperfect emission inventories, limited representativeness of point-wise observations and $0.1°$ grid cell, as well as differences between the ERA5 meteorology and the actual weather conditions at the campaign site.

**4 Idealised case of $NH_4NO_3$ fluxes**

The temporal evolution of a system where $NH_3$, $HNO_3$, and $NH_4NO_3$ interact to form an equilibrium has been studied in a single-column SILAM simulation. The simulation was initialised at a constant molar mixing ratio of $NH_4NO_3$ corresponding to a concentration of $5\mu gm^{-3}$. Fig. 2 shows the evolution of the vertical profiles of the mixing ratios for the three compounds, as well as their deposition rates and vertical fluxes at 2 m. The apparent deposition velocities of the species, i.e. their vertical

fluxes normalised with the corresponding concentrations, are plotted in Fig. 2f.

As seen in Fig. 2, the initial $NH_4NO_3$ immediately partly decomposes to form an altitude-dependent equilibrium with the gases $NH_3$ and $HNO_3$. The deposition fluxes of the gases are mostly controlled by their respective concentrations in the lowest layer, and gradually decrease as the surface layer becomes depleted. The deposition flux of sub-micron $NH_4NO_3$ is several orders of magnitude smaller than that of the gases. While the equilibrium allows for the existence of $NH_4NO_3$ at the

domain bottom, there is a substantial downward aerosol flux replenishing the decomposed amount in the lowest model layer. A remarkable upward flux of $HNO_3$ (panel e) is due to its somewhat slower deposition rate compared to $NH_3$ in the simulation. As a result, the disintegration of $NH_4NO_3$ (controlled by the $NH_3$ removal rate) produces more $HNO_3$ than becomes deposited. The excess $HNO_3$ diffuses upwards.





**Figure 2.** The temporal evolution of vertical profiles of concentrations of ammonium $NH_4NO_3$ (a), $NH_3$ (b), and $HNO_3$ (c) in the model experiment, and corresponding time series of the deposition fluxes at the surface (d), the downward fluxes at 2 m above the surface (e), and apparent deposition velocity at 2m (f).




The apparent deposition velocity, which would be reported by flux measurements at 2 m above the surface, is about $1\,\mathrm{cm\,s^{-1}}$
for most of the period, matching the one observed in many field campaigns used by Pleim et al. (2022). However, the simulations show that the apparent deposition velocity originates from the chemical decomposition of $NH_4NO_3$ and has no relation to the process of deposition of fine particles.

     Another remarkable feature of the simulated particle flux is that it is not proportional to the corresponding concentration. Indeed, once the $NH_4NO_3$ concentration drops (panels a-c), the apparent deposition velocity of $NH_4NO_3$ increases and,
theoretically speaking, approaches infinity at the moment when the $NH_4NO_3$ concentration tends toward zero (panel f). In reality, infinite apparent deposition velocities are not found in outdoor experiments due to the presence of other aerosols and finite gas-particle conversion rates.

     As long as $NH_4NO_3$ is present in the lowest layer, the fluxes of both gases are also heavily affected by the aerosol decomposition, which replenishes the gases while they are deposited. When $NH_4NO_3$ vanishes, so does the supporting mechanism that
kept the gaseous concentrations stable, and the downward fluxes of both gases become proportional to their concentrations. Therefore, the concept of deposition velocity is not only inapplicable to $NH_4NO_3$ fluxes, but also to $NH_3$ and $HNO_3$ fluxes, as long as $NH_4NO_3$ is present.

## 5   Discussion

In this section, we consider implications of the existence of non-conservative aerosols on the particle fluxes and apparent
deposition velocities.

### 5.1   Idealised systems with gas-particle equilibrium

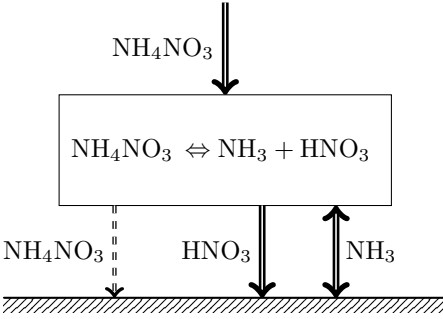

**Figure 3.** The process of ammonium nitrate deposition onto the surface. The thickness of the arrows qualitatively represents the magnitude of the flux. $NH_4NO_3$ flux to the surface appears only if the concentrations of gases at the surface allow for $NH_4NO_3$ existence

The deposition process described above is shown in Fig. 3. The downward flux of particles is controlled by gas-phase deposition at the surface rather than particle deposition. The resulting near-surface deficit of the gases is replenished via particle-to-gas conversion, which, in turn, creates a deficit of aerosols, leading to a net downward flux of ammonium nitrate.



Quite evidently, the presence of any semi-volatile substance with a high deposition rate of the gas phase will lead to the same result: the aerosol flux at a reference height will be controlled by gaseous deposition and particle-to-gas conversion.

While it is clear that gas-particle conversion breaks the proportionality between the fluxes and concentrations for both phases, the question arises: What concentration of a non-conservative aerosol species is needed to dominate the net particle downward flux? To answer this question, let us consider a mixture of $NH_4NO_3$ and an inert sub-micron aerosol with a deposition velocity

of $V_p = 0.001\,\mathrm{m/s}$ and at a concentration of $C_p$. The presence of ammonium nitrate implies that the concentrations of $NH_3$ and $HNO_3$ are sufficiently high for its formation, i.e. several $\mu g/m^3$, depending on the temperature. For such a system, the net downward particulate flux at the reference height will be a sum of the fluxes of the inert aerosol and ammonium nitrate:

$$F_{\mathrm{aer}} = F_{\mathrm{NH4NO3}} + F_p = \max(C_{\mathrm{HNO3,sat}}V_{\mathrm{HNO3}}, C_{\mathrm{NH3,sat}}V_{\mathrm{NH3}}) + C_p V_p \tag{4}$$

The apparent aerosol deposition velocity can be computed as a ratio of the downward flux and aerosol concentration. As-

suming that $NH_3$ is deposited slower than $HNO_3$, one obtains that

$$V_{\mathrm{aer,a}} = \frac{F_{\mathrm{aer}}}{C_{\mathrm{aer}}} = \frac{C_{\mathrm{HNO3,sat}}V_{\mathrm{HNO3}} + C_p V_p}{C_{\mathrm{NH4NO3}} + C_p} \tag{5}$$

If the ammonium nitrate concentration $C_{\mathrm{NH_4NO_3}}$ is small compared to $C_p$, the apparent deposition velocity for the total aerosols becomes simply:

$$V_{\mathrm{aer,a}} = V_p + V_{\mathrm{HNO3}}\frac{C_{\mathrm{HNO3,sat}}}{C_p}, \tag{6}$$

and the requirement of the small disturbance of the deposition velocity reads as:

$$C_p \gg C_{\mathrm{HNO3,sat}}\frac{V_{\mathrm{HNO3}}}{V_p} \tag{7}$$

This condition does not depend on $C_{\mathrm{NH_4NO_3}}$ (as long as it is small enough) and, since $V_{\mathrm{NHO3}} = 0.1\,\mathrm{m/s}$ and $V_p = 0.001\,\mathrm{m/s}$, leads to a requirement for the concentration of the inert aerosol $C_p \gg 100\mu g/m^3$. Such concentrations of inert aerosols are rare even in highly polluted locations. Therefore, in a vast majority of real-life cases, even a minor presence of ammonium nitrate

in the air guarantees that the apparent deposition velocity of particles is heavily affected by the particle-to-gas conversion near the ground.

## 5.2  Effect of finite gas-particle equilibration kinetics

The above calculations were performed assuming instant gas-particle conversion. The actual conversion rates depend on the features of the semi-volatile substance(s) and atmospheric conditions. Equilibration times have been considered by Meng and

Seinfeld (1996) for inorganic aerosols and by Shiraiwa and Seinfeld (2012) for organic aerosols. The main mechanism for such





equilibration is condensation/evaporation of material to/from the particle surface. The rate of this process is controlled by the diffusivity of the gases and by the availability of particle surface. The latter is a function of particle size and concentration.

Meng and Seinfeld (1996) concluded that for the considered cases of ambient environment, except for very low relative humidity conditions when particles stayed dry, the conversion of nitric acid to nitrate was completed within 1000 seconds. This time scale is probably relevant for most of ambient conditions, since there is strong evidence for ammonium-nitrate particles not crystallizing at a 10 – 20% relative humidity (Dougle et al., 1998), while in the simulations of Meng and Seinfeld (1996) deliquescence occurred at 50%.

For the validity of instantaneous gas-particle partitioning of organic compounds, Shiraiwa and Seinfeld (2012) conclude that it is clearly established "for relatively high volatility compounds partitioning into liquid particles, but ... breaks down for partitioning of low and semi-volatile compounds into liquid and semi-solid particles". For liquid particles the equilibration time scale is also quite short and approaches 1000 s only for particle sizes closer to $1\mu$m at PM concentrations below a few $\mu g/m^3$.

Using a 1000 s equilibration time as a conservative estimate, one can conclude that the vertical flux of particles with true deposition velocity of $0.1\,\mathrm{cm/s}$ is affected by the gas-particle conversion at a reference height of $1\,\mathrm{m}$, and detailed calculations are needed to account for it.

### 5.3 Intermittency, sign and scaling of particle fluxes

The intermittency of ambient particle fluxes can be clearly seen in the observational time series obtained with fast-response instruments, such as those reported by Gallagher et al. (1997) and many others (e.g Grönholm et al., 2009; Vong et al., 2004; Deventer et al., 2015; Petroff et al., 2018). One of plausible explanations is that the particle flux is shifting back and forth between the gas-phase- and particle-phase- driven a regimes due to changes in ammonia and nitrate concentrations, or changes in ambient temperature and humidity altering the gas-particle equilibrium.

In many cases, the observed fluxes of particles change their sign. The upward flux of particles can occur due to particle formation from the gas phase in the vicinity of the surface or if ammonia evaporates from the surface at a sufficient rate, enriching the near-surface layer with ammonium-nitrate particles (Nemitz et al., 2009).

So far, we have considered externally mixed particles, i.e., $NH_4NO_3$ or a conservative aerosol. In the presence of internally-mixed particles, that have a relatively stable core covered with ammonium nitrate staying in gas-particle equilibrium, opposite-sign fluxes of different particle sizes can occur. Larger particles, enriched with ammonium nitrate follow the surface flux of ammonia. If the gas concentration drops below its saturation limit, the ammonium nitrate shells evaporate, resulting in a shrinkage of the particle size and the formation of an opposite-sign flux of smaller particles. This mechanism explains the rather strong anti-correlation of particle fluxes of different sizes reported by Deventer et al. (2015) and Petroff et al. (2018).

Several field studies report a linear scaling of particle fluxes with friction velocity $u_*$, i. e. with the apparent deposition velocity being linear to it (Gallagher et al., 1997; Pryor, 2006; Contini et al., 2010; Lavi et al., 2013). This scaling is consistent with the assumption that particle fluxes are controlled by fast-depositing gases, whose deposition scales linearly with the turbulent diffusion in the surface layer.





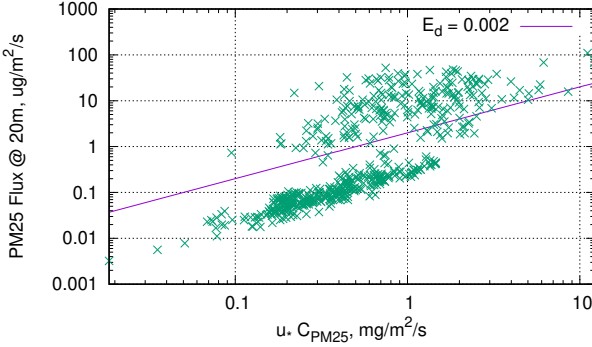

**Figure 4.** Hourly downwards particle fluxes vs concentrations multiplied with friction velocity, simulated for the Speulder site in the Netherlands for the month of June 1993, same simulation as in Fig. 1

The driving role of the gas-phase deposition is also consistent with the behavior of the apparent deposition velocity, normalized with $u_*$, as a function of atmospheric stratification. The minimum of the deposition velocity at a slightly-stable stratification has been observed by Wesely et al. (1985) and by (Gallagher et al., 1997; Lavi et al., 2013), with the shape of the function agreeing with the behavior of the stability function for scalar fluxes in the stratified surface layer.

### 5.4 Is the flux proportional to the concentration?

The vast majority of experimental studies on particle fluxes report the fluxes in terms of the apparent deposition velocity, without explicit verification of proportionality between the flux and the concentration. While the proportionality rather naturally holds for well-controlled wind-tunnel studies, it is not guaranteed for ambient particles in outdoor experiments.

To illustrate the non-trivial nature of this requirement, consider a scatter plot of $PM_{2.5}$ fluxes at a reference height vs the corresponding $PM_{2.5}$ concentrations (Fig 4), simulated by SILAM for the month of the (Gallagher et al., 1997) study (considered in Sec. 3). As a reference, we plotted a line corresponding to the dimensionless deposition velocity suggested by Wesely et al. (1985) for neutral and stable stratification, i.e.

$$E_d \equiv v_d/u_* = 0.002. \tag{8}$$

The points in the plot form two clearly distinguishable clusters corresponding to the two aforementioned regimes of the flux. The lower cluster corresponds to the flux of particles in the absence of ammonium nitrate, rendering them inert in the model. The cluster indicates a high correlation between the flux and the concentration. The residual scatter is caused by the different size distributions and water contents of the $PM_{2.5}$ components, as well as the varying meteorological conditions, in accordance to the SILAM deposition scheme (Kouznetsov and Sofiev, 2012). The upper cluster corresponds to the regime of gas-phase controlled particle flux, exhibiting much higher fluxes, a two orders of magnitude larger scatter of the values, and a much less pronounced dependency on the concentration.



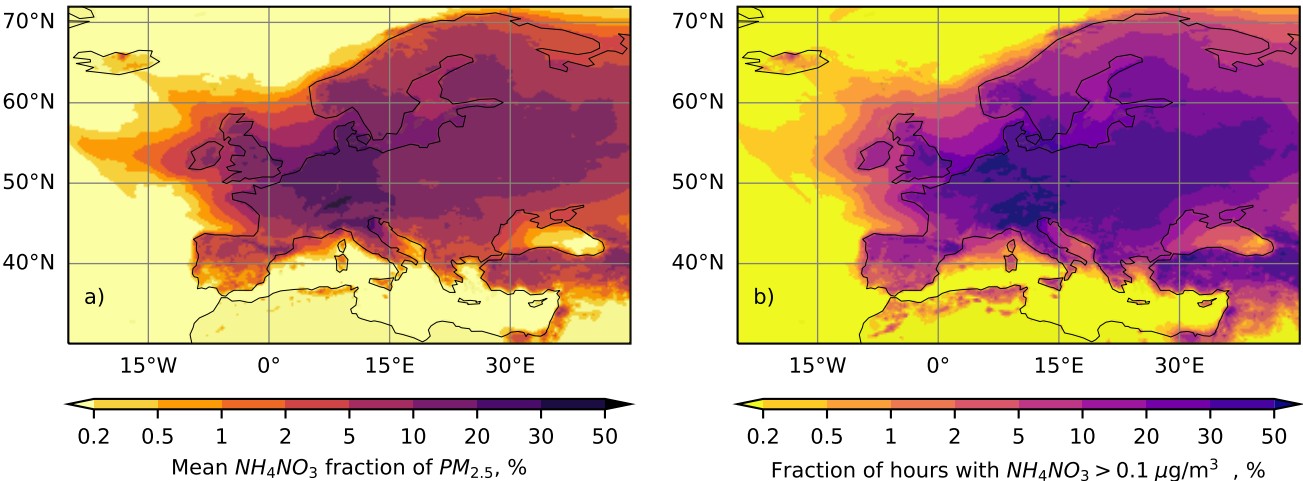

**Figure 5.** Mean fraction of $NH_4NO_3$ in $PM_{2.5}$ for the year of 2024 (a), and fraction of hourly average values with $NH_4NO_3$ exceeding $0.1\mu g/m^3$, according to SILAM forecasts made for the CAMS2-40 regional service

We have not found any study with such a scatter plot of the observed fluxes, or another way of examination of the proportionality between the flux and the concentration. However, time series that can be found in many studies (e.g Wesely et al., 1985; Petroff et al., 2018), suggest that the correlation between fluxes and concentrations is rather low in the outdoor studies.

### 5.5 Occurrence of ammonium-nitrate

To estimate the frequency of the occurrence of the near-surface ammonium nitrate, we analyzed the SILAM operational air quality forecasts for 2024, performed within the framework of the Copernicus CAMS2-40 regional operational service (Colette et al., 2024). The service consists of 11 European air-quality models that produce daily forecasts of atmospheric pollution at a spatial resolution of $0.1° \times 0.1°$ and at a temporal resolution of one hour. The forecasts have been extensively evaluated against multiple types of observational data. Copernicus publishes quarterly reports on the performance of each model (https://atmosphere.copernicus.eu/regional-services, accessed 15.4.2025), and has an extensive tool for online model evaluation (https://regional-evaluation.atmosphere.copernicus.eu/, accessed 15.4.2025). The service provides the by far most reliable forecasts for atmospheric pollution in Europe, and SILAM is among the best-performing models of the ensemble.

For the analysis, we chose the first day of the SILAM forecasts over the whole year and extracted the near-surface concentrations of $PM_{2.5}$ and $NH_4NO_3$. Figure 5a shows the mean relative contribution of ammonium-nitrate to the total $PM_{2.5}$ over the year. One can see that for most of the European territory, the fraction of ammonium-nitrate is within the range of 10–30%, whereas the mere presence of ammonium nitrate aerosol is an indicator of the gas-particle equilibrium strongly affecting the deposition fluxes. The frequency of such occurrences was estimated from the fraction of hourly $NH_4NO_3$ concentrations exceeding $0.1\,\mu g/m^3$ (Figure 5b). For the most part of the European territory, $NH_4NO_3$ aerosol is present in the air near the





surface for more than 30% of time. The particle fluxes over the Speulder forest site in the Netherlands, studied by Gallagher et al. (1997), are impacted by $NH_4NO_3$ for more than 50% of the time, at the Hyytiälä site in central Finland (Buzorius et al.,
2003; Pryor, 2006; Grönholm et al., 2007; Pryor et al., 2009; Rannik et al., 2009) — for more than 20% of the time, at the Waldstein site in Bavaria (Deventer et al., 2015) for over 50% of the time, and at the Yatir forest research station in Israel (Lavi et al., 2013) for about 10 to 20% of the time.

## 6   Conclusions

We identified a terminological inconsistency in the definition of the deposition velocity. For the modeling community, the
concept of deposition velocity implies a proportionality between the deposition flux and the atmospheric concentration, and all parameterizations of dry deposition velocity rely on this assumption. On the other hand, experimental studies define the deposition velocity as just a ratio of downward flux and concentration, formally calculated from the observed variables and being specific for each measurement. We have not found any experimental study where the proportionality between the fluxes and the concentrations was explicitly examined across the campaign. Time series available from some studies suggest that
correlation between fluxes and concentrations is rather low.

SILAM model simulations of both an idealized and real-life cases demonstrated that the vertical flux of particles is frequently not proportional to the particle concentration at a reference height. The proportionality is broken by the sink of particles between the measurement height and the surface, due to gas-particle transformation of ammonium nitrate.

The SILAM model, with the particle deposition scheme based on first principles and explicit gas-particle equilibration for
ammonium nitrate, is capable of reproducing observed fluxes of ambient aerosols that corresponds to velocity of $10^{-2} ms^{-1}$. However, the aerosol deposition velocity in the model is of the order of $10^{-4} ms^{-1}$. Moreover, the model reproduces the intermittent nature and temporal evolution of the observed particle fluxes.

We have shown that very low concentration of ammonium nitrate is sufficient to break the flux conservation, as it is broken rather by the presence of sufficiently high concentrations of nitric acid and ammonia, which trigger the gas-particle conversion.
Data on the abundance of ammonium nitrate over Europe from the CAMS2-40 European service suggest that ammonium nitrate conversion occurs over Europe for about half of the time.

In addition to ammonium nitrate, the presence of any semi-volatile substance may break the particle conservation near the surface following the same mechanism.

In laboratory experiments, the nature of the aerosol is usually well known, so the assumption of particle conservation holds.
In natural conditions, the assumption should be explicitly tested by evaluating whether the particle concentrations and corresponding fluxes are proportional to each other.

The presence of gas-particle conversion is sufficient to explain 100-fold differences between the apparent deposition velocities obtained in different experiments for similar surfaces and aerosol sizes.

The concept of deposition velocity (in a sense of Eq. 1) is a very powerful tool to reduce the complexity of dry deposition
in models. However, under ambient conditions, the presence of semi-volatile species breaks the aerosol conservation, and the

observed vertical fluxes cannot be directly used to infer the dry deposition velocity for models. Nevertheless, well documented observations of ambient fluxes provide invaluable means for the assessment of chemistry-transport models as a whole, when all relevant processes are considered, including gas-particle conversion near the surface.

*Code and data availability.* The code of SILAM model that can be used to reproduce the results of the current study is available from GitHub
https://github.com/fmidev/silam-model under the GPLv3 public license, which in particular allows for reusing the code in other models. The specific revision used for this paper has been archived at Zenodo (https://doi.org/10.5281/ZENODO.14608973, access 06.06.2025). The implementation of the particle dry deposition scheme (Kouznetsov and Sofiev, 2012) can be found from `depositions.silam.mod.f90` file, and implementation of ammonium-nitrate partitioning in `aerosol_dynamics_simple.f90`.

*Author contributions.* RK performed the numerical simulations and evaluations, wrote the initial text and prepared the figures. MS con-
tributed to the case conceptualization, participated in writing and editing of the manuscript. AU and RH participated in the model development and editing the manuscript.

*Competing interests.* The authors declare that no competing interests are present.

*Disclaimer.* The paper represents the authors' personal opinions and views, which might or might not agree with the positions of their organizations.

*Acknowledgements.* The study was supported by the European Union via Horizon Europe-funded CAMAERA project (No 101134927).



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
