# Peer review of "Deposition velocity concept does not apply to fluxes of ambient aerosols"

_EGUsphere, 2025_

## Author Comment (AC1)

**Response to RC1 (15.8.2025): Deposition velocity concept does not apply to fluxes of ambient aerosols**

Rostislav Kouznetsov[1], Mikhail Sofiev[1], Andreas Uppstu[1], and Risto Hänninen[1]

[1]Finnish Meteorological Institute, Helsinki, Finland

**Correspondence:** Rostislav Kouznetsov (Rostislav.Kouznetsov@fmi.fi)

*General impression*

*The authors revisit a problem that has been known for a while, i.e. that the dynamic gas-aerosol partitioning of between gas-phase ammonia and nitric acid one the one hand and particulate ammonium nitrate on the other causes fluxes that are non-conserved with height for the interacting compounds and particulate matter as a whole. I fully agree that most ambient*

5 *flux measurements are affected by this process and cannot and should not be used to derive the deposition velocity of particles at the ground, even at only small contributions from ammonium nitrate. Whilst it is worthwhile reiterating this message which is widely ignored, partially because it is difficult to deal with, unfortunately, this paper neither provides a full review of the subject nor does it add many new insights to the discussion already in the literature, provide a quantification of the impact of aerosol evaporation or offer a solution to the problem. The main new angle it adds, is framing the problem as a non-linearity*

10 *between flux and concentration. This I agree has not been done before, but I am also not sure it helps (see below).*

*In addition, there are some omissions, questionable assumptions and inconsistencies in the paper that would need to be addressed prior to publication.*

*The authors should take note of the study of (Ryder, 2010) which applied a more sophisticated model to the Speuld dataset, amongst others. I admit that this is unfortunately not very accessible via literature searches. Also not mentioned is the study*

15 *by Van Oss et al. (1998)who also simulated the Speuld dataset. There is a lot of other literature not cited in the paper (see also below), including a relatively recent study over forest (Katata et al., 2020).*

*Overall, I feel this paper is not publishable in its current form. It would either need to be significantly extended to become a more complete review of the subject or the analysis of the modelling would need to be expanded to provide more guidance of how to deal with the problem. In either case the modelling would need to be redone incorporating a kinetic constraint on the*

20 *attainment of the equilibrium (see below) and consideration of temperature gradients (if not already done).*

First of all we thank the reviewer for the time spent on reading the paper and for the very detailed and extensive comments. Here is a preliminary response to the main points. The points not covered here will be addressed in the revised paper and/or in the detailed response.

The message of the paper is indeed that the ambient particle fluxes are affected by gas-particle conversion and should not be used to derive the deposition velocity of particles. We would rather argue that this applies to any vertical flux of ambient accumulation-mode particles of detectable magnitude, but we (a) are not involved into the measurements and do not have strong evidence for this, and (b) believe that the proof of the applicability is a duty of those who use these data to infer particle deposition to surfaces.

The paper is not intended to be a literature review about the effects of aerosol evaporation on particle deposition, though indeed a short part on that should be added. There has been reviews of particle-flux measurements and deposition velocities derived from them, published as separate studies (e.g. Farmer et al., 2021), or as parts of other studies (e.g Pleim et al., 2022), and we did not aim to repeat them. Neither did we aim to provide a sophisticated simulation of the Speud dataset or any other outdoor experiment. Detailed size- and chemistry- resolved modelling studies about it have already been published (Ryder, 2010; Van Oss et al., 1998; Katata et al., 2020), and we thank the reviewer for pointing out that.

Our goal is much more specific and tailored for GMD as a model development journal, i.e. to address the two-orders of magnitude discrepancies between the aerosol deposition velocities derived from first principles and those calculated from the field observations. The practical goal is, roughly speaking, to illustrate that performing linear regression (with slope $V_d$) through the cloud of points on a plot (see e.g. Fig. 4 of Pleim et al., 2022), does not lead to a reasonable result for ambient aerosols, and that using such a fit as an "effective model" is not sensible, as more accurate models can be implemented. This is essential for regional and global-scale modelling of particle dispersion.

To achieve our goal, we have illustrated the problem in the modelling realm, and show that the reported "high" apparent deposition velocities can be perfectly reproduced with the model that implements the first-principles based scheme and suggests a "low" deposition velocity for sub-micron particles. Thus we do not agree with the referee in that we are not providing a solution, i.e. a model implementing a first principles based deposition aerosol velocitiy and simple particle-gas conversion is significantly better for reproducing measurement results than an effective $V_d$ model, and also easily implementable in a regional or global model.

For this purpose, a simple model with a single particle bin and instant equilibrium is the simplest and the most-transparent one that can still reproduce the behavior of apparent deposition velocity. A more complicated setup, certainly available in the SILAM model, would only overshadow the effect.

We also need a clear criterion to distinguish between a deposition velocity that can be used in parameterizations of the form Eq. 1, and those that cannot. To our surprise, the criterion of the proportionality between flux and concentration, despite being seemingly trivial, probably has not been formulated before, as admitted by the reviewer.

We agree, however, that the issue of non-instant conversion of NH4NO3 deserves more attention than we gave in the paper. We will include new simulations of the single-column case with coarsened vertical resolution and with/without conversion kinetics. We do not expect it to change the bulk surface fluxes of ammonia and nitrates much, but changes in the near-surface profiles and fluxes may, indeed, be interesting to discuss.

*Major comments:*

*1. I am not sure the modelling and measurement communities have different definitions for Vd. The modelling community uses the form F = (-) Vd x C to derive the flux from the modelled concentration, the measurement community uses the equation Vd = (-) F/C to derive Vd from flux measurements, often for use in models. This Vd is meant to be the same, that is usually the rationale for making the measurements. There are a number of reasons why the flux measured at a certain height above the surface does not represent the processes at the surface, such as advection, storage and, for non-inert compounds, chemical conversion. These errors need to be corrected for or shown to be negligible. The measurement community measuring fluxes of NH3, HNO3, NH4+ or NO3- is getting increasingly aware of this, but the community measuring particle number fluxes less so. I do not think that this is due to a difference in definition; it is a problem of awareness and lack of solutions to deal with the problem.*

We agree that the naming scheme of the different communities is probably confusing. There are multiple observational papers which plainly report "deposition velocities" (actually apparent ones), and there are multiple modelling papers that apply these values to derive a $V_d$, used as a direct parameter in Eq. (1), which leads to the discrepancy. We will rephrase the sentence in the revised manuscript.

We will not make a-priori assumptions on what was meant beyond the content of the published papers. If one wants to use simultaneous flux and concentrations observations to identify $V_d$ as a parameter of the proportionality model in Eq. (1), there are well established procedures for that (although hitherto seemingly not applied within this field). One should make a regression, accounting for error bars, check if the specific linear model is suitable for the specific data set, etc. It should be made separately for each set of conditions. This procedure does not involve calculating the ratios of individual fluxes to the corresponding concentrations, as it is not a statistically correct procedure. We are not aware of any experimental paper where such a processing has been done for fluxes and concentrations of ambient particles. Therefore, we can conclude that the proportionality has probably not been assumed as a requirement or, at least, did not receive due attention in these studies.

*2. The authors specifically comment on Nemitz et al. (2009) explicitly mentioning the problem, but still calculating Vd. This is no different in this paper which also deals with the problem and then calculates a deposition velocity (Fig. 2f), and refers to the measured deposition velocities as Vds (Fig. 1). This entity is explained in the caption, but (as a symbol) also has a clearly defined meaning from which this paper diverges. It is difficult to get away from these inconsistencies. Papers that are aware of the problem often use the term "apparent deposition velocity" or "local deposition velocity". I agree that maybe a different symbol is needed by the community to distinguish this from the Vd that describes the deposition process at the ground. Here the authors use subscript "a" as an option. What might be more important, however, is that the associated text correctly describes how to interpret the presented values of Vd and what they really represent.*

Thank you! We will double-check consistency.

*3. The paper is not framed in the context of what the models are being used for. Total aerosol deposition is not normally a metric of interest. The main interest is in surface PM2.5 concentrations (for health impacts) and in total nitrogen deposition (for ecosystem impacts). For the former, deposition is considered a loss process for the PM2.5 (and/or its chemical components) from the atmosphere. For this purpose it is of secondary importance whether the removal of the ammonium nitrate is due to*

*actual deposition to the surface or evaporation, and the concept of an "apparent" or "effective" deposition velocity may*
*suffice and is arguably the appropriate loss term to use (except for the fact that the non-mechanistic treatment of the process*
*then does not allow for tracking of the gas phase compounds and their potential to reform NH4NO3 further aloft). If the goal*
95 *is to quantify nitrogen deposition, the evaporation matters and as does the fate of the ammonia and nitric acid. How much of*
*it is deposited and how much is transported upwards (line 188) and how can this be estimated?*

We agree that aerosol deposition itself is not a metric of interest for some applications, but we disagree that it is an excuse to simulate it wrongly.

From a model development standpoint, surface PM2.5 can be "tuned" by offsetting the error in $V_d$ by tweaking other parts
100 of the model. The orders-of-magnitude discrepancies between the $V_d$ schemes mentioned in the first paragraph are usually not visible in the surface PM skill scores, as all AQ models that implement these schemes have been tuned to reproduce measured PM2.5 concentrations. However, reproducing PM concentrations through error compensation is not a correct approach.

Eutrophication and acidification problems indeed require the deposited amounts of nitrogen regardless of its gaseous or particular phase. An "effective" deposition velocity can be tuned to obtain roughly correct total deposition amounts at a specific
105 site, for example in the Netherlands, but this approach would certainly break down under substantially different conditions, since the proportionality enforced by using the deposition velocity approach does not hold either for particles or for gases in the presence of particles, as explained at the end of Sec 4. Neither can one formulate a reasonable deposition velocity for substances like "total oxidized nitrogen" due to the same lack of proportionality. However, as in the case of PM2.5 concentrations, obtaining locally reasonable total deposition amounts can be achieved by applying an "effective" deposition velocity for both
110 species, since the total depositions are somewhat constrained by the emissions, and the models can be tuned by multiple means to obtain a reasonable deposition range (i.e. by offsetting an error with other errors).

Unlike argued by the referee, accurate dry deposition of particles is important for multiple scenarios covered by transport models, e.g. long-range transport of particular pollutants and deposition of hazardous substances in particular form, such as radioactive fallout.

115 To obtain reasonable deposition amounts of nitrogen in ambient conditions, one does not need any particle deposition scheme, as long as gas-particle conversion and gaseous deposition are handled reasonably. If particle-deposition is the only mechanism of NH4NO3 removal in a model, the effective deposition velocity will be wrong anyway, as the the downward flux of nitrogen is not a function of the NH4NO3 concentration. However, for the modelling of e.g. radioactive fallout, a two-order-of-magnitude discrepancy in the deposition velocity for fine particles is hardly acceptable.

120 We will add a paragraph on the cases when accurate particle-surface interaction is important.

To quantify deposition vs upward transport of ammonia and nitric acid one would need detailed information about in-soil processes of ammonia (a.k.a. compensation point) and a detailed description of the behaviour of deposited nitric acid in the soil, which are probably larger sources of uncertainty than the specific details of particle-surface interaction.

*4. The paper needs to distinguish from the start between two related but different effects: the impact of aerosol evaporation*
125 *on (a) bulk deposition fluxes, e.g. of nitrate mass or total PM mass, and (b) the impact on size segregated particle flux mea-*

*surements measured over a specified size bin. The authors acknowledge the second effect in the paragraph starting in line 260, but ignore it in their attempt to reproduce the fluxes measured at Speuld (Fig. 1), which is impacted by exactly that effect as previously shown (Ryder, 2010). In this context, I am not convinced the (size-segregated) Speuld dataset can be reproduced with a bulk model. It needs a model that explicitly simulates the change in the particle size distribution.*

We fully agree that details of the size-segregated dataset can not be reproduced with a bulk (non-size segregated) model, neither did we attempt to do that. Our goal is to address the problem of particle deposition in large-scale chemistry-transport models that neither can afford detailed particle size information nor have sufficient input to constrain it. However, bulk models can be strikingly good in catching the overall effect, if aerosol evaporation is included. The temporal variability of the apparent deposition velocity $V_{d,a}$, shown in Fig. 1 of the paper and in Fig. 1 of (Gallagher et al., 1997), has been reproduced with a simple bulk model and instant gas-particle equilibrium. We will highlight this in the revised paper.

*5. The analysis in the paper assumes chemical equilibrium to be instantaneous, but concedes that this is probably not the case and later discuss potential kinetic constraints and time-scales. This is critical. If there were no constraint on evaporation then there would never be any NH4NO3 near the ground where concentrations of NH3 and HNO3 converge towards 0. Without a kinetic constraint the effect is overestimated. The authors omit to mention that kinetic constraints have been used in virtually all other model studies of this effect, either deploying rate constants (Kramm and Dlugi, 1994) or a relaxation time scale towards equilibrium (Brost et al., 1988; Van Oss et al., 1998; Nemitz and Sutton, 2004, e.g.) So why is this neither used here? At a time scale of 1000s evaporation likely starts to become unimportant given that transport timescales are smaller; e.g. see timescale analysis of Wolff et al. (2010) In fact, equilibrium is not usually maintained even at the measurement height with concentrations affected by conditions further aloft (e.g. Aan De Brugh et al., 2012)*

None of these simulations deal with particles in general, which is the primary interest of this paper. However, we agree that additional discussion of the topic is relevant and we will add it, also including simulations with kinetics included.

*6. From a modelling perspective, the problem is that (a) aerosol evaporation is an unresolved subgrid process in atmospheric chemistry and transport models which do not have the vertical resolution to calculate the thermodynamic partitioning according to the vertical concentration and temperature (and RH) gradients near the ground and (b) these models assume aforementioned equilibrium. Maybe it should be stated in these terms.*

This is rather a matter of resources and efficiency of implementation. One can explicitly account for kinetics of evaporation for vertical profiles inside and above the canopy in limited-scale simulations used for research, where computational costs are of secondary importance. However, applying an effective $V_d$ concept for such particles would be wrong at any scale, regardless the model complexity.

*7. The paper identifies a problem, but does not offer a solution. If the deposition concept is not applicable, what should be used in its stead? For example, the EMEP model acknowledges aerosol evaporation as a non-resolved process within the model and uses a parameterisation of an "effective deposition velocity" for the constituents of ammonium and nitrate (Simpson et al., 2012; Eq. (68) and associated text). This has been developed by fit to nitrate flux measurements (apparent deposition*

*velocity). This is a first approximation that should be improved with time. In terms of nitrogen deposition it implicitly makes*
160 *the assumption that the ammonia and nitric acid that are liberated during evaporation are in fact deposited.*

We disagree with this. SILAM is an operational model which has successfully implemented the solution without excessive costs.

Generally speaking, why would one need a deposition velocity for ambient aerosol? In the modelling realm one normally knows the properties of the simulated compounds, and can choose a suitable deposition scheme for them, such as via a depo-
165 sition velocity, a prescribed flux, no deposition at all or something else.

For inert particles, we recommend using schemes deduced from first principles and laboratory measurements involving controlled conditions and known aerosols. An example of this is the SILAM aerosol dry deposition scheme (Kouznetsov and Sofiev, 2012). Evaporating/condensing particles should be treated by explicitly accounting for these processes. This is more accurate even in a simplified form than any scheme involving an "effective" deposition velocity.

170 *8. The text suggests that the reduction in gas-phase concentrations towards the ground, caused by their deposition (and in particular of HNO3; line 166), is the main driver of evaporation. The model results of (Ryder, 2010) suggest that at Speuld this effect is only responsible for about half of the evaporation, the remainder being driven by increases in temperature (and equilibrium vapour pressures) near the ground and inside the canopy. In other words, even if NH3 and HNO3 did not deposit at all there would still be a driver for evaporation. Clearly, this varies over the day as the temperature gradients change. Are*
175 *the temperature gradient accounted for in the model?*

SILAM does account for the temperature and humidity dependence of NH4NO3 saturation, but not for in-canopy temperature and humidity profiles. More detailed canopy-resolving models are needed for the latter. However, if NH3 and HNO3 did not deposit at all, their concentration profiles would vary with height and time driven by equilibrium with NH4NO3. This would not cause any additional vertical fluxes.

180 *9. Whilst I fully agree that the evaporation complicates the relationship between the surface (and also the measured) flux and concentration, field observations are unlikely to show a linear relationship for a number of reasons. Vd is not a simple function of u* (as implied by some of the parameterisations including that reflected in Fig. 4), but also atmospheric stability (as discussed in the paper), size distribution, surface roughness (which might change with footprint and wind direction) etc. In addition, measurement uncertainty complicates any relationship. As the paper correctly describes, the traditional deposition*
185 *velocity concept relies on fluxes being constant with height and therefore does not apply to reactive compounds. Because Vd is effectively the slope between flux and concentration, it automatically follows that the relationship also becomes less meaningful and proportionality breaks down. I am not sure this is an independent observation / statement.*

Yes, this pretty much describes the situation. Indeed, there are various ways to handle the observation uncertainty, atmospheric stability, footprint differences, etc. However, if one wants to apply experimental results to constrain the function in
190 Eq. 1, $V_d$ should be determined as the slope valid only for those very specific conditions. As a by-product one can obtain the uncertainty of the slope and information on the applicability of $V_d$ for the considered case. Otherwise, the resulting $V_d$ is just a ratio of two weakly connected parameters, both having noticeable uncertainties.

If the proportionality is not reproducible in the experiments, enforcing the measured ratio of flux to concentration in a chemistry-transport model as an effective deposition velocity is obviously not a good choice.

195 We will add the clarification in the revised manuscript.

We thank the reviewer for the detailed and constructive comments and are looking forward to prepare a more extensive response and a revised version of the manuscript. The specific comments will be also addressed in the final response.

**References**

Aan De Brugh, J. M. J., Henzing, J. S., Schaap, M., Morgan, W. T., Van Heerwaarden, C. C., Weijers, E. P., Coe, H., and Krol, M. C.: Modelling the Partitioning of Ammonium Nitrate in the Convective Boundary Layer, Atmos. Chem. Phys., 12, 3005–3023, https://doi.org/10.5194/acp-12-3005-2012, 2012.

Brost, R. A., Delany, A. C., and Huebert, B. J.: Numerical Modeling of Concentrations and Fluxes of HNO3, NH3, and NH4NO3 near the Surface, Journal of Geophysical Research, 93, 7137–7152, https://doi.org/10.1029/JD093iD06p07137, 1988.

Farmer, D. K., Boedicker, E. K., and DeBolt, H. M.: Dry Deposition of Atmospheric Aerosols: Approaches, Observations, and Mechanisms, Annu. Rev. Phys. Chem., 72, 375–397, https://doi.org/10.1146/annurev-physchem-090519-034936, 2021.

Gallagher, M. W., Beswick, K. M., Duyzer, J., Westrate, H., Choularton, T. W., and Hummelshøj, P.: Measurements of Aerosol Fluxes to Speulder Forest Using a Micrometeorological Technique, Atmospheric Environment, 31, 359–373, https://doi.org/10.1016/S1352-2310(96)00057-X, 1997.

Katata, G., Matsuda, K., Sorimachi, A., Kajino, M., and Takagi, K.: Effects of Aerosol Dynamics and Gas–Particle Conversion on Dry Deposition of Inorganic Reactive Nitrogen in a Temperate Forest, Atmos. Chem. Phys., 20, 4933–4949, https://doi.org/10.5194/acp-20-4933-2020, 2020.

Kouznetsov, R. and Sofiev, M.: A Methodology for Evaluation of Vertical Dispersion and Dry Deposition of Atmospheric Aerosols, Journal of Geophysical Research, 117, D01 202, 1–17, https://doi.org/10.1029/2011JD016366, 2012.

Kramm, G. and Dlugi, R.: Modelling of the Vertical Fluxes of Nitric Acid, Ammonia, and Ammonium Nitrate, J Atmos Chem, 18, 319–357, https://doi.org/10.1007/BF00712450, 1994.

Nemitz, E. and Sutton, M. A.: Gas-Particle Interactions above a Dutch Heathland: III. Modelling the Influence of the $NH_3$-$HNO_3$-$NH_4NO_3$ Equilibrium on Size-Segregated Particle Fluxes, Atmospheric Chemistry and Physics, 4, 1025–1045, https://doi.org/10.5194/acp-4-1025-2004, 2004.

Nemitz, E., Dorsey, J. R., Flynn, M. J., Gallagher, M. W., Hensen, A., Erisman, J.-W., Owen, S. M., Dämmgen, U., and Sutton, M. A.: Aerosol Fluxes and Particle Growth above Managed Grassland, Biogeosciences, 6, 1627–1645, https://doi.org/10.5194/bg-6-1627-2009, 2009.

Pleim, J. E., Ran, L., Saylor, R. D., Willison, J., and Binkowski, F. S.: A New Aerosol Dry Deposition Model for Air Quality and Climate Modeling, J Adv Model Earth Syst, 14, e2022MS003 050, https://doi.org/10.1029/2022MS003050, 2022.

Ryder, J.: Emission, Deposition and Chemical Conversion of Atmospheric Trace Substances in and above Vegetation Canopies, School for Earth, Atmospheric and Environmental Sciences University of Manchester, Manchester, 2010.

Van Oss, R., Duyzer, J., and Wyers, P.: The Influence of Gas-to-Particle Conversion on Measurements of Ammonia Exchange over Forest, Atmospheric Environment, 32, 465–471, https://doi.org/10.1016/S1352-2310(97)00280-X, 1998.

Wolff, V., Trebs, I., Foken, T., and Meixner, F. X.: Exchange of Reactive Nitrogen Compounds: Concentrations and Fluxes of Total Ammonium and Total Nitrate above a Spruce Canopy, Biogeosciences, 7, 1729–1744, https://doi.org/10.5194/bg-7-1729-2010, 2010.

---

## Author Comment (AC2)

**Response to RC2 (9.9.2025): Deposition velocity concept does not apply to fluxes of ambient aerosols**

Rostislav Kouznetsov[1], Mikhail Sofiev[1], Andreas Uppstu[1], and Risto Hänninen[1]

[1]Finnish Meteorological Institute, Helsinki, Finland

**Correspondence:** Rostislav Kouznetsov (Rostislav.Kouznetsov@fmi.fi)

*The authors have done a good job formulating, demonstrating and discussing the issue, but unfortunately there is rather little practical outcome and recommendations on how to deal with the identified problems (given the paper estimates quite an abundance of ammonium nitrate in Europe and indicates its potentially frequent presence in PM2.5?*

*Could the authors suggest better solutions for the application of Vd in the case of semi-volatile (ammonium nitrate) aerosols*
5 *i.e. whether and how the "apparent Vd" from field measurements could be used in a sound way? Alternatively, could a model representation of the apparent Vd be proposed? Actually the paper anticipates that the authors would (L. 105) "..suggest an approach to bridge the gap between observed apparent deposition velocities and deposition parametrisations based on Eq. (1)".*

In practice, we recommend using deposition schemes designed for passive particles only for passive particles. For semi-
10 volatile particles, the particle deposition scheme should be applied in combination with gas–particle partitioning. This approach reproduces observed fluxes without invoking apparent Vd. The Apparent Vd should therefore be treated as a diagnostic quantity only, not as a tunable parameter of the model. We will make this recommendation more explicit in the revised paper.

*It would also be advisable to show the effect of accounting for gas-aerosol partitioning during aerosol deposition on model results on a regional (European) scale. What is the seasonality of this effect? Does it help to improve the model performance.*

15 The gas–aerosol partitioning plays a major role for semi-volatile species, but it is hardly possible to isolate the effect of partitioning during deposition from other processes occurring throughout the aerosols/precursors life cycle (emissions, chemical transformation, and transport). A simulation with partitioning entirely turned off can be done with moderate effort. However, the results will have no meaning because the formation of ammonium nitrate is the result of the very same partitioning process that is responsible for its disintegration near the surface. So, in the simulation with partitioning off, there will be no ammonium
20 nitrate in aerosols at all.

*My recommendation would be that the paper can be published after the authors complete the manuscript in line with the suggestions in the evaluation above and also attend to the following comments.*

Thank you! The specific comments will be addressed in the revised manuscript.